# Embryonic Stem Cell-Derived Neurons as a Model System for Epigenome Maturation during Development

**DOI:** 10.3390/genes14050957

**Published:** 2023-04-22

**Authors:** Sally Martin, Daniel Poppe, Nelly Olova, Conor O’Leary, Elena Ivanova, Jahnvi Pflueger, Jennifer Dechka, Rebecca K. Simmons, Helen M. Cooper, Wolf Reik, Ryan Lister, Ernst J. Wolvetang

**Affiliations:** 1Australian Institute for Bioengineering and Nanotechnology, The University of Queensland, Brisbane, QLD 4072, Australia; 2School of Biomedical Sciences, The University of Queensland, Brisbane, QLD 4072, Australia; 3Australian Research Council Centre of Excellence in Plant Energy Biology, School of Molecular Sciences, The University of Western Australia, Perth, WA 6009, Australia; 4Harry Perkins Institute of Medical Research, Perth, WA 6009, Australia; 5Epigenetics ISP, The Babraham Institute, Cambridge CB22 3AT, UK; 6Queensland Brain Institute, The University of Queensland, Brisbane, QLD 4072, Australia; 7The Wellcome Trust Sanger Institute, Hinxton CB10 1SA, UK

**Keywords:** epigenomics, neuroscience, DNA methylation, neuronal maturation, cell culture systems

## Abstract

DNA methylation in neurons is directly linked to neuronal genome regulation and maturation. Unlike other tissues, vertebrate neurons accumulate high levels of atypical DNA methylation in the CH sequence context (mCH) during early postnatal brain development. Here, we investigate to what extent neurons derived in vitro from both mouse and human pluripotent stem cells recapitulate in vivo DNA methylation patterns. While human ESC-derived neurons did not accumulate mCH in either 2D culture or 3D organoid models even after prolonged culture, cortical neurons derived from mouse ESCs acquired in vivo levels of mCH over a similar time period in both primary neuron cultures and in vivo development. mESC-derived neuron mCH deposition was coincident with a transient increase in Dnmt3a, preceded by the postmitotic marker Rbfox3 (NeuN), was enriched at the nuclear lamina, and negatively correlated with gene expression. We further found that methylation patterning subtly differed between in vitro mES-derived and in vivo neurons, suggesting the involvement of additional noncell autonomous processes. Our findings show that mouse ESC-derived neurons, in contrast to those of humans, can recapitulate the unique DNA methylation landscape of adult neurons in vitro over experimentally tractable timeframes, which allows their use as a model system to study epigenome maturation over development.

## 1. Introduction

The unique epigenomic landscape of neurons is hypothesized to allow these postmitotic cells to respond to diverse environmental stimuli during development and to modify gene transcription in response to activity while retaining their cellular identity [1,2,3,4,5,6]. DNA methylation is thought to play an important role in imparting this simultaneous robustness and adaptability to neurons [7,8,9]. In most somatic cells, DNA methylation is largely restricted to cytosines in the context of CG dinucleotides (mCG). The methylation of CG sites is considered a relatively stable modification with a well-described function in gene silencing and imprinting [10]. In contrast, in adult mammalian brains, other types of DNA methylation are found at high levels, including non-CG methylation (mCH; where H = A, T, or C [11,12,13,14] and intermediates in the DNA demethylation pathway, particularly 5-hydroxymethylcytosine (5hmC) [13,15,16,17]. In adult human and mouse neurons up to ~50% of methylated cytosines in the genome occur in the CH dinucleotide context, a level similar to mCG [11,13], and the majority of this exists in the CA dinucleotide sequence context. While the precise roles of these modifications are not fully understood, the complex and diverse methylation profiles of adult neurons [18] suggest that DNA methylation plays an important role in the dynamic and adaptable regulation of gene expression in these cells. Studies in both mice and humans have shown that mCA is first observed in the brain shortly after birth and continues to accumulate during development to adulthood, after which the levels remain stable [13]. This observation raises the exciting possibility that the generation of mCA may link early life experiences with neuron function later in life. The level of intragenic mCA in neurons inversely correlates with transcript abundance [6,14,19] and in mice mCA deposition, is negatively regulated by gene transcription [6], suggesting that mCA functions as a part of a molecular system that modulates gene expression in response to synaptic activity and that consolidates the specificity of neuron subtypes.

DNA methylation is established and maintained by a family of conserved DNA (cytosine-5)-methyltransferases (DNMTs). Dnmt1 propagates existing methylation patterns at symmetrically opposed CG sites during cell division and is essential for the maintenance of methylation and chromosomal stability [7,20]. Dnmt3a and Dnmt3b, on the other hand, catalyse the de novo methylation of cytosine, and the levels of Dnmt3a can be dynamically regulated to increase DNA methylation in the brain [21]. The postnatal deposition of mCH is driven by a transient increase in the expression of Dnmt3a [6,11,13,22,23], and the conditional deletion of Dnmt3a in Nestin-positive neuronal precursors during late gestation results in impaired motor activity [22]. In contrast, deletion of Dnmt3a in excitatory neurons at early the postnatal stages was reported to have no apparent major effect on brain development or function [3], suggesting that the developmental window during which mCH is deposited is critical [13]. The importance of DNA methylation in governing correct neuronal function is exemplified by a range of developmental neurological disorders that result from mutations in proteins associated with DNA methylation in both the CG and CH context [23,24].

Defining the roles of mCG and mCH in neuron maturation and synaptic plasticity is of fundamental importance for understanding normal and abnormal brain development, and thus a tractable and representative in vitro model system to further explore this process is highly desirable. We therefore investigated the levels, distribution, and temporal dynamics of mCH during in vitro neuronal differentiation of human and mouse pluripotent stem cells. Deploying a range of cellular, genomic and transcriptomic assays, we discovered similar subnuclear patterning, levels, and spatiotemporal dynamics of DNA methylome reconfiguration during the differentiation and maturation of mouse neurons, in vitro and in vivo, that inversely correlate with gene expression levels, but we also differences that likely arise from the complex influence of the in vivo cellular environment on neuron differentiation and maturity.

## 2. Materials and Methods

### 2.1. Reagents and Antibodies

The primary antibodies are detailed in Appendix A. Directly conjugated Alexa488-mouse anti-NeuN was purchased from Millipore (MAB377X). Mouse monoclonal antibodies to 5-methylcytosine-adenosine dinucleoside (mCA) and 5-methylcytosine-guanine dinucleoside (mCG) were raised against KLH-conjugated dinucleosides at the Technology Development Laboratory (Babraham Bioscience Technologies Ltd., Cambridge, UK; anti-mCA antibody clone 2C8H8A6 currently available from Active Motif, Cat. No 61783/4). Alexa Fluor secondary antibodies were purchased from Life Technologies. ESGRO recombinant mouse Leukaemia Inhibitory Factor (LIF) was obtained from Merck Millipore. The remaining reagents were obtained from Thermo Fisher Scientific or Sigma Aldrich unless otherwise specified.

### 2.2. Mouse Tissue Collection and In Vitro Neuron Differentiation

Mice were obtained from the UQBR mouse breeding facility. Adult C57BL/6 mice (8–10 weeks old) were sacrificed via CO_2_ asphyxia, mice were perfused with PBS, and brains were dissected and snap frozen on dry ice. Brains were then sectioned on a cryostat at 12 microns, placed on poly-L-lysine slides (VWR), and stored at −80 °C until processing. All experimental procedures were approved by the Animal Welfare, Experimentation and Ethics Committee at the Babraham Institute and were performed under licenses by the Home Office (London, UK) in accordance with the Animals Scientific Procedures Act 1986.

For primary neurons, hippocampal or cortical neurons were cultured from embryonic-day-18 C57BL/6 mouse brains as described previously [25]. All procedures were conducted according to protocols and guidelines approved by the University of Queensland Animal Ethics Committee. Isolated E18 neural progenitors were either frozen directly for whole genome bisulfite sequencing (WGBS) or plated onto 0.1 mg/mL of Poly L-Lysine/8 µg/mL laminin-coated plates in Neurobasal containing 2% B27, 0.5 mM of L-glutamine, and 1% Pen-Strep and maintained for 14 days. 14 DIV neurons were either dissociated with Accutase and frozen as cell pellets for WGBS or washed in PBS and fixed in 2% PFA/PBS for ICC.

### 2.3. Murine Embryonic Stem Cell Culture and In Vitro Neuron Differentiation

Two mESC cells lines, R1 (ATCC SCRC-1011), derived from a 129X1 × 129S1 male blastocyst [26], and G4, derived from a 129S6/SvEvTac × C57BL/6Ncr male blastocyst [27], were maintained on gamma-irradiated mouse embryonic fibroblasts in mESC medium (Dulbecco’s Modified Eagle Medium (DMEM; GIBCO), 15% Hyclone-defined FBS (GE Healthcare, Chicago, IL, USA), 103U/mL of ESGRO LIF (Millipore, Burlington, MA, USA), 1× L-Glutamax, 1× sodium pyruvate, NEAA 1× (Invitrogen, Waltham, MA, USA), 0.1 of mM beta-mercaptoethanol). Cells were fed daily and split based on confluency. As the differentiation of the G4 mESC line was slightly more robust, this was used routinely, and the data shown refer to G4 ESC-derived neurons unless otherwise stated.

Neural differentiation was initiated by dissociating the mESC colonies using Accutase, and excess feeder cells were removed from the cell suspensions by panning on gelatine-coated plates. Dissociated cells were counted and transferred to ultra-low attachment cell culture plates at a dilution of 220,000 cells/mL in differentiation medium (Dulbecco’s Modified Eagle Medium (DMEM; GIBCO), 10% Hyclone-defined FBS (GE Healthcare), 1× L-Glutamax, 1× sodium pyruvate, 1× NEAA (GIBCO), and 0.1 mM of beta-mercaptoethanol). Neural induction was continued for 8 days, and medium was changed every two days and supplemented with 5 µM of retinoic acid for the final 4 days. Cell aggregates were dissociated using Accutase, and single cells were selected through a 40 µm cell strainer and plated onto 0.1 mg/mL poly-ornithine/4 µg/mL laminin-coated plates in N2 medium (DMEM/F12, 1× Glutamax 1× N2 supplement, 20 µg/mL of insulin, 50 µg/mL of BSA) at a density of 50,000–100,000 cells/cm^2^. After 2 days, cells were changed into N2B27 (Neurobasal, 1× N2 supplement, 1× B27 supplement, 1× Glutamax). N2B27 was changed every 4 days for 12 days, after which 50% media changes were performed for the remaining culture time.

High potassium-mediated depolarisation was performed as previously described [28]. Briefly, cells were washed with low K+ buffer (15 mM of HEPES, 145 mM of NaCl, 5.6 mM of KCl, 2.2 mM of CaCl_2_, 0.5 mM of MgCl_2_, 5.6 mM of D-glucose, 0.5 mM of ascorbic acid, and 0.1% BSA at pH 7.4), transferred to high K+ buffer for 5 min (formulated as the low K+ buffer but with 95 mM of NaCl and 56 mM of KCl), washed again in low K+ buffer, and either harvested directly for RNA, immediately fixed for ICC, or chased in growth medium for the appropriate times prior to fixation or RNA extraction.

### 2.4. Human In Vitro Neuron Differentiation

Human neurons were generated following two different approaches: either as adherent monolayer cultures via a neural stem cell intermediate population [29] or as cerebral organoids [30]. For adherent cultures, PSCs were disaggregated mechanically and cultivated as embryoid bodies in DMEM/F12 and Neurobasal 1:1 mix, 1% Glutamax (all Life Technologies) with 1:200 N2 Supplement (R&D Systems, Minneapolis, MN, USA), 1:100 B27 supplement without retinoic acid (Miltenyi Biotec, Bergisch Gladbach, Germany) supplemented with 10 µM of SB-431542, 1 µM of dorsomorphin (both Selleckchem, Houston, TX, USA), 3 µM of CHIR 99021 (Cayman Chemical Company, Ann Arbor, MI, USA) and 0.5 of µM Purmorphamine (Sigma, St. Louis, MO, USA) for 3 days on petri dishes. On day 4, SB-431542 and dorsomorphin were removed and 150 µM of ascorbic acid (Cayman Chemical Company) was added to the media. On day 6, cells were plated onto Matrigel- (Corning, Tewksbury, MA, USA) coated TC dishes to allow attachment of neuroepithelial cell types. Over several passages, neural stem cells were enriched, resulting in pure populations. These neural stem cells could be differentiated by removing the small molecules from the media and switching to B27 supplement with retinoic acid, resulting in a mixed population of beta3-Tubulin-positive neurons and GFAP-positive glia within 4 weeks. For cerebral organoids, PSCs were dissociated into single cells and plated in an ultra-low-attachment 96-well plate at 9000 cells/well in E8 media (Life Technologies, Carlsbad, CA, USA) with 50 µM of Y-27632 (Selleckchem, Houston, TX, USA). After 5–6 days, embryoid bodies were transferred to ultra-low-attachment 6-well TC plates in neural induction media (DMEM/F12, 1% Glutamax, 1% nonessential amino acids and 10 µg/mL of heparin (Sigma, St. Louis, MO, USA). After another 4–5 days, organoids were embedded into Cultrex (R&D Systems) matrix and cultivated under continuous agitation on an orbital shaker in cerebral organoid media (DMEM/F12 and Neurobasal 1:1 mix, 1% Glutamax, 0.5% nonessential amino acids, 1:200 N2 supplement, 1:100 B27 supplement (Miltenyi Biotec, Bergisch Gladbach, Germany), and 1:40,000 insulin (Sigma, St. Louis, MO, USA), with media changes being performed every 3 days until a maximum age of 9 months.

### 2.5. RT-qPCR

Total RNA was extracted using the Macherey-Nagel NucleoSpin RNA kit (cat. no. 740955.50), including on column digestion of DNA with RNase-free Dnase according to the manufacturer’s specifications. Concentration and 260/280 ratios were quantified using a NanoDrop 1000 spectrophotometer before cDNA synthesis using the iScript cDNA synthesis kit (Bio-Rad, Hercules, CA, USA) was performed. Primers were designed to span exon–exon boundaries wherever possible (Appendix A). When this was not possible, samples were excluded if genomic DNA contamination was more than 10-fold over the cDNA concentration. Quantitative PCR (qPCR) reactions were conducted using SsoFast Evagreen (Bio-Rad) with a cDNA template as well as C1000 Thermocycler (Bio-Rad, Hercules, CA, USA) and CFX software according to the manufacturers’ instructions. Results were analysed as described previously [31].

### 2.6. Immunocytochemistry

As described previously [32], protein ICC was performed on cells fixed in 4% paraformaldehyde or 2% paraformaldehyde (Electron Microscopy Sciences, Hatfield, PA, USA). For double and triple immunocytochemistry using antibodies to methylated DNA, a sequential labelling method was used. Briefly, cells were fixed in 2% PFA/PBS for 10–30 min, permeabilised for 1 h in 0.5% TX-100, and then depleted of residual methylated RNA using Rnase A at 10 µg/mL for 30 min at 37 °C in PBS. Cells were subsequently blocked in PBS/0.2% BSA/0.2% cold-water fish skin gelatine for 10 min and then labelled with protein-targeting antibodies in PBS/0.2% BSA/0.2% cold-water fish skin gelatine overnight at 4 °C. Cells were then washed with PBS, labelled using corresponding secondary antibodies in PBS/0.2% BSA/0.2% cold-water fish skin gelatine for 30 min, washed in PBS, and then refixed in 2% PFA/PBS for 15 min. DNA epitopes were retrieved using 4N HCl/0.1% TX-100 for 10 min at room temp, and cells washed in PBS/0.05% Tween-20 and then blocked in PBS/0.05% Tween-20/1% BSA (BS) for 1 h. Antibodies to methylated DNA were applied in PBS/0.05% Tween-20/1% BSA overnight at 4 °C. Cells were subsequently washed in PBS/0.05% Tween-20/1% BSA, and appropriate secondary antibodies were applied in the same buffer containing 0.1 µg/mL of DAPI for 30 min^−1^ h. Finally, cells were washed in PBS/0.05% Tween-20/1% BSA, rinsed in PBS, and either imaged directly (high-throughput imaging), or mounted in Mowiol (Sigma, St. Louis, MO, USA).

### 2.7. Immunohistochemistry

Mouse brain sections were fixed in 2% PFA for 30 min, then permeabilised in PBS/0.5% Triton X-100 for 1 h, blocked in PBS with 1% BSA (blocking solution, BS) for 1 h and incubated overnight at 4 °C with a NeuN antibody (1:2500) in BS. They were subsequently incubated for 1 h with a secondary fluorescently labelled antibody (Alexa488, Alexa568, Alexa647; Thermo Fisher, Waltham, MA, USA) in BS (1:1000). After post-fixation with 2% PFA for 10 min, to obtain access to DNA methyl groups, cell nuclei were mildly depurinised with 4N HCl treatment for 15 min and incubated with antibody-enriched culture medium against mCG or mCA in BS overnight at 4 °C. After incubation for 1 h with a secondary fluorescently labelled antibody in BS (1:1000) the tissue was stained with 1:100 YOYO1 (Thermo Fisher, Waltham, MA, USA) for 15 min and mounted with Vectashield Antifade Mounting Medium (H-1000, Vector Laboratories, Newark, NJ, USA). Washing with PBS/0.05% Tween-20 for 1–2 h was performed after each step, reagents were dissolved in PBS/0.05% Tween-20 and steps performed at room temperature, unless otherwise stated.

### 2.8. Optical Microscopy

Phase contrast microscopy was performed on live cells using an Olympus IX51. Confocal microscopy was performed on fixed cells using either a Zeiss 710 confocal microscope and a 40× water immersion objective or a Leica SP8 confocal microscope using a 60× oil immersion objective. High-throughput imaging was performed using a Perkin-Elmer Operetta equipped with a 20× air objective. Brain section imaging was performed with a Nikon A1-R confocal microscope and a 60× 1.4 NA oil immersion objective.

Image analysis was performed via the manual masking of nuclei and measuring fluorescence intensity within the masked area (integrated density) (Figure 1E,F) or with in line scans (Figure 2B) using Image J or by high-throughput Operetta image acquisition using Harmony (Figure 1D, Appendix A) and analysis of fluorescence intensity using Cell Profiler [33]. For normalisation of mCA labelling, data were acquired using identical acquisition settings prior to pooling and normalisation. For each individual replicate, a minimum of 50 nuclei were analysed with the exception of those presented in Figure 2B, with 10 nuclei being analysed. All image analysis data were collated using Excel for Office 365, and graphs were prepared using Excel for Office 365 or Graphpad Prism 8.

### 2.9. Transmission Electron Microscopy

ESC-derived neurons were incubated with 10 µg/mL of CTB-HRP in either high K+ or low K+ buffer for 5 min, washed in PBS, and fixed in 2.5% glutaraldehyde (Electron Microscopy Sciences, Hatfield, PA, USA) for 24 h. Following fixation, cells were processed for 3, 39-diaminobenzidine (DAB) cytochemistry using standard protocols. Fixed cells were contrasted with 1% osmium tetroxide and 4% uranyl acetate prior to dehydration and embedding in LX-112 resin [32]. Sections (~50 nm) were cut using an ultramicrotome (UC64; Leica, Wetzlar, Germany). To determine CTB-HRP endocytosis, presynaptic regions were visualised at 60,000× using a transmission electron microscope (model 1011; JEOL, Akishima, Japan) equipped with a Morada cooled CCD camera and iTEM AnalySIS software, version 1.0.

### 2.10. Whole Genome Bisulfite Sequencing

DNA methylation analysis with WGBS was performed using 100–500 ng of genomic DNA, isolated with the DNeasy Blood and Tissue Kit (Qiagen, Hilden, Germany) with some modifications: samples were incubated for 4 h at 56 °C with an additional RNAse A incubation for 30 min at 37 °C. Following this, 500 ng of genomic DNA spiked with 4% (*w*/*w*) unmethylated lambda phage DNA (Thermo Fisher Waltham, MA, USA was sheared to a mean length of 200 bp using the Covaris S220 focused-ultrasonicator. Libraries for WGBS were prepared as follows: DNA fragments were end-repaired using the End-It kit (Epicentre, Madison, WI, USA), A-tailed with Klenow exo- (NEB, Ipswich, MA, USA), and ligated to methylated Illumina TruSeq adapters (BIOO Scientific, Austin, TX, USA) with DNA Ligase (NEB), which was followed by bisulfite conversion using the EZ DNA-methylation Gold kit (Zymo Research, Irvine, CA, USA). Library fragments were then subjected to 7 cycles of PCR amplification with KAPA HiFi Uracil+ DNA polymerase (KAPA Biosystems, Wilmington, MA, USA). Sequencing was performed either with single-end 100 bp on a HiSeq 1500 or a MiSeq (Illumina, San Diego, CA, USA) or paired-end 2 × 150 bp on a NovaSeq 6000 (Illumina, San DIego, CA, USA).

### 2.11. DNA Methylation Analysis

Reads were trimmed for quality and adapter sequences removed. Following preprocessing, reads were aligned to mm10 or hg19 references with Bowtie and a pipeline described previously [34,35], resulting in a table summarising methylated and unmethylated read counts for each covered cytosine position in the genome. Bisulfite nonconversion frequency was calculated as the percentage of cytosine base calls at reference cytosine positions in the unmethylated lambda control genome. This was performed individually for each context (CA, CC, CG, CT). Methylation for particular genomic contexts and the average methylation of gene bodies were calculated by intersecting whole genome data with feature bed files from the UCSC table browser using Bedtools [36]. For correlation, heatmap generation, and clustering of methylation data, Deeptools was used [37], along with the R packages pals, gplots, ggplot2, viridis, and RColorBrewer. During the calculation of the average methylation level in a region (weighted methylation level), the number of C basecalls divided by the total sequence coverage at reference C positions was used to calculate the methylation level for each context (CG, CH, CA, CC, CT). Sample-specific and context-specific bisulfite nonconversion rates, calculated from the unmethylated lambda phage DNA control, were subtracted for each methylation context. Gene set enrichment analysis was performed using the fgsea package in R [38] with reactome [39] and gene ontology annotations [40,41]. Pathways were sorted by the NES score (enrichment score normalised to mean enrichment of random samples of the same size), and only pathways with *p*-value < 0.05 were considered in subsequent analyses.

For similarity analysis, every gene was given a score as follows: First, the difference in weighted DNA methylation between in vivo and in vitro neurons (y) was calculated for each gene and scaled to a value between 0 and 1 (using the formula x = |y − 1|) so that genes that were more similar in methylation state between both samples would have a value (x) closer to 1. Then, the average of weighted DNA methylation per gene in neuronal samples was compared against the average weighted DNA methylation of the glial and fetal samples, resulting again in a score between 0 and 1, with genes showing greater differences having a value closer to 1. Both scores for similarity between neurons and dissimilarity to non-neuronal samples were added in order to give both aspects the same weight and were scaled to values between 0 and 1, with values closer to 1 representing genes being more similar in methylation state between both neuronal samples but different compared to the glia and fetal frontal cortex. This similarity score was then used to rank all genes for GSEA with the fgsea package as described above.

DMR calling was performed using the HOME algorithm [42] with standard pairwise comparisons between in vivo and in vitro neurons separately for CG and CH contexts. DMRs were then filtered for spanning at least 3 Cs with a minimum coverage of 5 per sample, having more than 3 Cs per 100 bp and a minimum methylation difference of 0.2. Filtered DMRs were intersected with genomic context bed files to call numbers of DMRs within them. Enrichment was calculated by dividing counted numbers by expected numbers based on the size of individual features compared to the total genome.

Enrichment of transcription binding factors in enhancer elements was performed using homer2 v. 4.11 [43] by using those enhancers with a defined methylation difference between neuron groups against the background of all enhancers of the dataset using the findMotifs method.

### 2.12. RNA-seq

RNA was extracted via a hot Trizol extraction method using 100,000 NeuN-positive sorted nuclei per sample. Nuclei were washed in PBS and resuspended in Trizol (Sigma) at 65 °C and shaken at 1300 rpm for 15 min. RNA was enriched with a guanidinium HCl buffer and silica-coated magnetic beads with a DNAse I treatment step. RNA amounts and quality were assessed on a TapeStation using RNA Screen Tape (Agilent, Santa Clara, CA, USA), and 20–100 ng of total RNA was used per replicate to generate RNA-seq libraries. ERCC ExFold RNA Spike-In mixes (Thermo Scientific, Waltham, MA, USA) were added as the internal control. Libraries were prepared using the TruSeq Stranded mRNA library prep kit (Illumina, San Diego, CA,, USA) using TruSeq RNA unique dual index adapters (Illumina, San Diego, CA, USA). Libraries were quantified with qPCR on a CFX96/C1000 cycler (Biorad, Hercules, CA, USA) and sequenced on a Novaseq 6000 (Illumina, San Diego, CA, USA) for 2 × 53 bp as paired-end.

Fastq files were aligned to the mm10 transcriptome using HISAT2 v. 2.1.0 [44], and the resulting bam files were sorted and indexed using SAMtools v. 1.9 [45]. BPM normalization was performed with the deepTools v. 3.3.1 bamCoverage function [37] while GTF files for TPM values were calculated with stringtie v. 1.3.3b [46].

### 2.13. Fluorescence-Activated Nuclear Sorting

Intact nuclei were isolated from cell pellets as described previously [47,48]. Briefly, cells were dounce homogenised on ice in chilled nuclear extraction buffer (10 mM of Tris-HCl, pH 8, 0.32 M sucrose, 5 mM of CaCl2, 3 mM of Mg(Ac)2, 0.1 mM of EDTA, 1 mM of DTT, 1× protease inhibitor cocktail (Merck, Darmstadt, Germany), 0.3% Triton X-100). Nuclear lysates were filtered (40 µm), centrifuged for 7 min at 3000 rpm at 4 °C, and resuspended in PBS. Nuclei were blocked with 10% normal goat serum and labelled for 60 min on ice with either directly conjugated mouse anti-NeuN-Alexa488, preconjugated rabbit anti-Nanog/goat anti-rabbit Alexa488, or rabbit anti-Pax6/goat anti-rabbit Alexa488 complexes. Samples of each nuclear fraction were retained for secondary only antibody controls. 7-AAD (20 µg/mL) was added to all samples 15 min prior to sorting. A BD Influx cell sorter was used to sort nuclei. Prior to sorting, the secondary only control was used to gate events to isolate nuclei from cell debris. From the selected nuclear populations, nuclei were separated into distinct NeuN +ve/7-AAD +ve and NeuN −ve/7-AAD +ve populations or Nanog +ve/7-AAD +ve and Pax6 +ve/7-AAD +ve populations depending upon the cell type. For human 2-D cultures, no NeuN expression was detected after 6 weeks of culture. To sort for neuronal nuclei from these samples, the INTACT method was used to express a GFP fluorescence tag linked to nuclear envelope protein Sun1 driven by the neuronal promoter MAP2 [49].

### 2.14. Statistical Analysis

All data were analysed using an unpaired, two-tailed Student’s *t*-test unless stated otherwise. Additional information is given in respective figure legends. Bioinformatic analysis employed statistical tests as defined in the used software packages, with additional information given in the technical documentation of those software packages.

## 3. Results

### 3.1. Immunocytochemical Labelling for mCA Accumulated in Postmitotic Neurons and Temporally Correlated with DNMT3a Expression

In vivo, mouse cortical neurons begin to acquire readily detectable levels of mCH around 2 weeks after birth, which continues to increase up to 6 weeks of age and remains high throughout adulthood [13]. To assess whether this can be recapitulated in vitro, we used two independent approaches. First, we isolated primary cortical and hippocampal neurons from day-18-embryonic C57BL/6 mice (E18, average gestation 18.5 days) and cultured these for 14 days in vitro (DIV). We hypothesized that if mCH accumulation is due to cell intrinsic developmentally hardwired processes, 14 DIV should correlate with the temporal acquisition of mCH in vivo (Figure 1A). In the second approach, we adapted an established differentiation protocol to generate mouse cortical neurons from embryonic stem cells (mESCs) [50]. We hypothesized that if this developmental model recapitulates neural development and neuronal maturation in vivo, mCH accumulation would occur within several weeks (Figure 1B). Two different mouse ESC lines, R1 [26] and G4 [27], were differentiated as cell aggregates for 8 days in suspension, which was followed by dissociation and continued differentiation in adherent culture for up to 30 additional days to yield mixed cultures enriched in postmitotic neurons (Appendix A). Both cell lines developed mature neurons with an equivalent time course and to a similar extent, as assessed by morphology and immunohistochemistry (R1 and G4), TEM analysis of synaptic depolarization (R1), and c-Fos mRNA and ICC analysis following depolarization (G4; Appendix A, [51]). To investigate the temporal acquisition, subnuclear localisation, and cell-type specificity of mCH in cultured neurons, we used an antibody raised against the mCA dinucleotide (anti-mCA) to analyse the primary- and mESC-derived neuronal cultures with immunocytochemistry. To identify neurons, cultures were co-labelled for NeuN/Rbfox3, a well-established marker of most postmitotic neuron subtypes, and beta3-tubulin (Tubb3), a pan-neuronal marker. Specificity of the mCA antibody was confirmed using a panel of competitive methylated oligonucleotides (Figure 1C and Appendix A).

To first determine whether in vitro cultured neurons could acquire CH methylation, 14 DIV cortical and hippocampal primary neuronal cultures were immunolabelled for mCA, NeuN, and TUBB3 (Figure 1C). At this DIV, the majority of cells displayed a strong intranuclear labelling for mCA in addition to labelling for both neuronal markers. Nuclear labelling for mCA was completely abrogated by the competitive [5mC]A oligonucleotide (Figure 1C and Appendix A), confirming the specificity of the antibody.

We next used the anti-mCA antibody to analyse the accumulation of mCA at different times up to 38 days during the differentiation and maturation of mESC-derived neurons (Figure 1D). Neuronal cells were labelled for Tubb3 one day after attachment (day 9, Appendix A), and detectable NeuN labelling was observed within 3–6 days of attachment (differentiation day 11–14, Figure 1D), suggesting the rapid development of a postmitotic phenotype. Interestingly, this temporally correlates with the initial identification of NeuN at E10.5 in the embryonic mouse brain [52], suggesting that these developmental milestones are temporarily hardwired and can be recapitulated in vitro. Consistent with the early development of postmitotic neurons, there was no further increase in the number of NeuN-positive neurons over the 4 weeks in adherent culture although there was an obvious but highly variable increase in the number of non-neuronal cells within this time, including of glial cells (see Figure 1F), as described previously [53]. Immunocytochemical (ICC)-based analysis of DNA methylation in NeuN-positive cells revealed an increase in the level of nuclear mCA labelling between days 18 and 28, which remained high to day 38 (Figure 1D). As the initial observation of mCA was significantly later than was the initial observation of NeuN, a postmitotic phenotype is likely a prerequisite for subsequent acquisition of mCA. Consistent with this, we found only minimal labelling for mCA in Pax6-positive neural progenitors differentiated for 9–10 days relative to ~2-fold higher levels in NeuN-positive neurons differentiated for 28–38 days (Figure 1E). Similarly, we found minimal labelling for mCA in GFAP-positive glial cells and in additional unidentified cell types within the cultures that did not label for either neuronal or glial markers (Figure 1F).

Methylation of CH sites has been shown to be catalysed by Dnmt3a in vivo [6,21]. We therefore analysed the transcript abundance of Rbfox3 (NeuN) and the DNA methyltransferases Dnmt3a and Dnmt1 during differentiation with RT-qPCR (Figure 1G). Consistent with the results of the ICC, Rbfox3 (NeuN) expression was found to significantly increase between days 9 and 18, reaching a plateau at around day 28. In contrast but in agreement with previous data from both primary neurons and mouse brain [13,21], Dnmt3a expression transiently increased between days 18 and 28 of differentiation and decreased again by 38 days. This transient increase in Dnmt3a transcript level temporally correlated with the accumulation of mCA labelling observed b they ICC in the neurons (Figure 1G), strongly supporting our prediction that Dnmt3a catalyses CA methylation in the mESC-derived neurons. Dnmt1 expression was initially high (Day 9) but rapidly decreased after cell attachment, which points to its role in the maintenance of DNA methylation in the mitotic neural precursors. Collectively, these data demonstrate that mCA deposition occurs specifically in postmitotic neurons but not in non-neuronal cell types and correlates with DNMT3a expression.

### 3.2. mCA Labelling Was Enriched at the Nuclear Periphery in Neurons

During our ICC analysis of mCA labelling (Figure 1), we observed that immunolabelling for this DNA modification appeared enriched near the nuclear periphery in our in vitro neuronal cultures. To directly examine the distribution of DNA methylated in specific sequence contexts, neurons were labelled for either mCA, mCG, or 5-hydroxymethylcytosine (5hmC), which is also highly enriched in neuronal genomic DNA [15,16,17] (Figure 2A). This labelling confirmed that mCA was enriched at the nuclear periphery, suggesting an association with the nuclear lamina. Quantification of the intranuclear mCA labelling pattern using line scans across a randomly chosen subset of nuclei showed a clear enrichment towards the nuclear periphery that was not observed for NeuN (Figure 2B). In contrast to mCA, neither mCG nor 5hmC displayed strong peripheral labelling. mCG labelling was observed in highly labelled foci in addition to a low diffuse labelling throughout the nucleus. 5hmC was observed in diffuse patches within the nucleus consistent with an enrichment in euchromatin [54]. While both of these structures could be detected at the nuclear periphery, neither showed the strong preferential peripheral localisation observed for mCA.

To further examine the difference in intranuclear labelling for mCG and mCA, we then took advantage of an antibody to total methylated cytosines (5mC) that could label both mCA and mCG. Analysis of the triple-labelled mESC-derived neurons (mCA, 5mC, and DAPI) showed a partially overlapping labelling pattern (Figure 2C) that could be clearly demonstrated using line scans of the labelled nuclei. While mCA labelling was enriched at the nuclear periphery, total 5mC labelling was distributed between the nuclear periphery and intranuclear foci. These foci stained strongly for DAPI, which was consistent with condensed DNA, and can be assumed to relate to the strong mCG labelled foci shown in Figure 2A.

Finally, to confirm that the distribution of mCA observed in vitro faithfully represented the intranuclear localisation in vivo, we analysed the localisation of mCA and mCG in immunohistochemical sections of the adult mouse brain (Figure 2D). Both cortical and hippocampal brain sections showed an enrichment of mCA labelling at the nuclear periphery and enrichment of mCG labelling in intense intranuclear foci. Collectively, these data show that DNA methylated in the CA context has a unique intranuclear distribution as assessed with immunocytochemistry and shows a preferential localisation to the nuclear periphery, which is consistent with a potential association with the nuclear lamina in neurons.

### 3.3. mESC-Derived Neuronal Cultures Acquired mCH to Levels Similar to Those Observed In Vivo

Having shown that both mESC-derived neurons and cultured primary neurons acquired mCA labelling using immunocytochemistry, we next determined the global levels of DNA methylation (CG and CH) by WGBS.

In primary neurons, we found very low levels of mCH (<0.1%, weighted DNA methylation level: mCH/CH) in E18 in cells isolated from either the hippocampus or the cortex, which is consistent with previous in vivo data [13] and E12.5 mouse NPCs cultured in vitro [55]. Following 14 DIV, this level increased to 0.70% and 0.79% in the primary cortical and hippocampal neurons, respectively (Figure 3D), confirming that the mechanisms underpinning accumulation of mCH are conserved in in vitro cultures. This is in agreement with the in vitro differentiation of E12.5-derived mouse NPCs, which also showed an increase in mCH levels over several weeks of culture, reaching a maximum of 0.35% after 21 days [55]. Analysis of the mCH context confirmed that methylation at CA sites was the most abundant modification although smaller increases in methylation in the CT and CC contexts were also observed (Figure 3E–G). Comparison of the methylation levels at 14 DIV to those of the 2-week-old mouse prefrontal cortex (PFC, [13]) showed that these were similar for all three mCH subtypes. In contrast, the in vitro levels of mCG were slightly higher than the in vivo levels (Figure 3C).

Analysis of the global DNA methylation levels with WGBS was then undertaken on the mESC-derived neuronal cultures between days 27 and 38 postdifferentiation, and these periods were temporally comparable to approximately 1 week and 3 weeks postnatal in vivo development, respectively (Figure 3A,B). Consistent with the primary neuron analysis, both the G4- and R1-derived neuronal cultures acquired high levels of mCH within 27 days, consisting predominantly of mCA. The global level of mCH and mCA was very similar between the two cell lines and showed a time-dependent increase up to 38 days. Global levels of mCA and mCH at 38 days (equivalent to ~3 weeks postnatal in vivo) closely mirrored those of the prefrontal cortex of 2-week-old mice (Figure 3D,E), although the levels at 27 days (~1 week post-natal in vivo) were ~5-fold higher than those in 1-week old-mouse PFC. Whether this reflects an earlier deposition of mCH in vitro or is a result of differing proportions of neuronal and non-neuronal cells in the two sample types is not known. Smaller increases in the level of mCT and mCC were also observed in vitro, as was an increase in the level of mCG. Our data indicate that in vitro neuronal differentiation of mESCs recapitulates the overall in vivo levels of mCH and mCA, which is distinct from a prior report in transdifferentiated induced neuronal (iN) cells where much lower levels of mCH and mCA were observed [55]. Thus, we conclude that acquisition of mCH and mCA is largely a neuro-developmentally hardwired process.

### 3.4. Global mCH and mCG Levels in mESC-Derived Neurons Revealed Hypermethylation Relative to In Vivo Adult Neurons

As mESC-derived neuronal cultures contain multiple cell types (Figure 1), fluorescence-activated nuclei sorting (FANS) was used to analyse mCG and mCH levels in specific populations of cells (Figure 4A,B and Appendix A). To analyse the developmental timeline of mESC-derived neurons we sorted Nanog-positive nuclei from mESCs, Pax6-positive nuclei from mESC-derived neuronal progenitor cells, NeuN-positive nuclei from cells cultured for 30 days, and NeuN-positive and NeuN-negative nuclei from cells cultured for 38 days (Figure 4A, see also Figure 1). To investigate a possible temporal regulation of methylation patterns, we also isolated neuronal nuclei from various human PSC-derived neuronal cultures, including both 2-D cortical differentiation [29] and 3-D cerebral organoids [30] (Figure 4B). We then used WGBS to analyse the levels of mCG and mCH in these different nuclear populations, representing different cell types within the various neuronal differentiation timelines (Figure 4A,B).

During the mouse ESC differentiation process, global CG methylation levels were observed to progressively increase, with the highest increment observed during the maturation of Pax6-positive neural progenitors to 30-day old neuronal nuclei, corresponding to the period during which maximal Dnmt3a transcript abundance was observed in bulk cultures (Figure 1G). The observed levels of mCG in NeuN-positive nuclei at both 30 days and 38 days was substantially higher than that reported in NeuN-positive nuclei isolated from mouse PFC and was higher than the equivalent NeuN-negative cells in the cultures, suggesting that the hypermethylation of neuronal CG occurs in vitro. Interestingly, hypermethylation of CG was not observed in human cultures, where global levels of mCG were generally lower than those in vivo (for four out of five cultures).

We next analysed the levels of mCH in the various nuclei populations. In the mouse samples, the level of mCH was found to increase substantially during the transition from Pax6-positive NPCs to 30-day-old NeuN-positive neurons and again between day 30 and day 38 of neuron maturation (Figure 4D). The levels attained by day 38 of culture exceeded those of the 7-week-old mouse prefrontal cortex (approximately 65 days total development from the blastocyst stage), suggesting that for mCH and mCG, hypermethylation of the in vitro-derived neurons was occurring. This pattern of methylation was recapitulated for the individual analyses of mCA, mCC, and mCT (Figure 4E–G). For mCA, we found a level of 1.25% (mCA/CA) in Nanog-positive mESC-derived nuclei, which was consistent with published studies in mice [56,57] and humans [58,59]. This level was found to decrease in Pax6-positive neural progenitors and subsequently increase to 2.9% (day 30) and 5.3% (day 38) in NeuN-positive nuclei. As with mCG and mCH, at day 38 this level was higher than that reported in NeuN-positive nuclei isolated from mouse brains, again suggesting hypermethylation of CA. Consistent with the ICC results and published in vivo data [13], NeuN-negative cells within the 38-day old neuronal cultures contained low levels of mCA. The levels of mCT in the day 38 ESC-derived neurons reached levels similar to those in the adult mouse brain, while only negligible levels of mCC were detected in any cell type. Together these data demonstrate that mESC-derived neurons acquire non-CG methylation levels similar to in vivo levels.

In contrast to mouse ESC-derived neurons, only negligible levels of non-CG methylation were observed in any of the human ESC-derived neuronal populations, suggesting that even temporally extended human neuronal cultures are unable to mature sufficiently to acquire mCH. No difference in mCH was detected between shorter, 2-D neuronal cultures and aged cerebral organoids, suggesting that the culture conditions alone do not promote the acquisition of mCH.

### 3.5. Genome-Wide Distribution of mCH and mCG between In Vivo and mESC-Derived Neurons Indicated Regional Hypermethylation

To establish the degree to which DNA methylation patterns in the ESC-derived neurons recapitulated those of in vivo neurons, we generated base resolution methylomes using WGBS and assessed the regional distribution of mCG (Figure 5) and mCH (Figure 6). An average coverage per cytosine of around 30× was achieved across 3 replicates. In addition, we used nuclear RNA-seq to determine the relationship between DNA methylation and gene expression in NeuN-positive ESC-derived neurons. We then compared this to previously published datasets from 7-week-old mouse PFC glial cells (glia) or NeuN-positive neurons (in vivo neurons) and the fetal mouse frontal cortex (fetal) [13].

The analysis of the average mCG levels across all gene bodies and associated 10 kb flanking regions (Figure 5A) showed that these were generally similar between fetal, glial, and in vivo adult neurons but exhibited a generalised increase in ESC-derived neurons (designated as in vitro neurons) (Figure 5A, top plots) consistent with the observed global mCG levels (Figure 4). Analysis of mCH levels (Figure 6A) showed that these were very similar between in vivo and in vitro neurons and were much higher than either that of the glial or fetal samples (Figure 6A, top plot). No significant relationship between gene length and level of methylation was observed for mCG and only for the very largest genes for mCH (Appendix A). Genes were subsequently ordered by difference in mean gene body methylation level in vitro relative to in vivo (Figure 5A and Figure 6A, lower heatmaps), and the transcript abundance for each gene was determined (gold/purple heatmaps). In this analysis, we observed that 38.7% of the genes were mCG hypermethylated (gene body ΔmCG > 0.1) for in vitro neurons compared to for vivo neurons, while only 0.6% of the genes showed mCG hypomethylation (ΔmCG > 0.1). In the CH context, the ratio was more balanced, with 16.0% of the genes being hypermethylated (gene body ΔmCH > 0.01) and 12.2% of the genes being hypomethylated (ΔmCH > 0.01). Thus, while global and average mCG and mCH gene methylation levels were similar between the in vitro and in vivo neurons, differences in the level of gene body DNA methylation were evident. In both methylation contexts, gene body methylation appeared to inversely correlate with gene expression (see also Figure 7). As a recent study described DNA methylation patterns in directly reprogrammed mouse neurons (iN cells, [55]), we also compared our gene body methylation data to the iN cell data. This analysis showed that the hypermethylation of mCG observed in mESC-derived neurons was unique to this developmental model, as both the levels and the patterns of mCG observed in iN cells were similar to those of the in vivo neurons (Appendix A). However, as the overall level of gene body mCH/CH in iN cells was very low, hampering a direct comparison, we additionally compensated for the lower overall mCH levels by normalising gene body methylation levels to the average global methylation level within the different neuron populations. For mCG, despite the slightly lower methylation levels in the iN cells, when methylation patterns were normalised, a higher degree of similarity was observed to the mESC-derived neurons, suggesting that neither in vitro neuron population attains genic mCG patterning identical to in vivo neurons. For mCH, more complex differences were observed, with both iN cells and ESC-derived neurons showing regional variation in the degree of similarity to in vivo neurons.

To investigate the relationship between CG and CH methylation within gene bodies in the ESC-derived neurons, we assessed mCH patterns over all genes ordered by gene body mCG difference and gene body mCG patterns over all genes ordered by gene body mCH difference (Appendix A).

These analyses revealed that the patterns of DNA methylation levels observed were different for each context (compared to Figure 5A and Figure 6A). Indeed, a very low Pearson correlation of differences in methylation for both contexts between ESC-derived neurons and in vivo neurons (r = 0.0688) suggests that independent processes define DNA methylation levels and that individual genes are mCG- and mCH-methylated to a different degree compared to the global average levels. Plotting the total number of calls at cytosine reference positions within gene bodies in the same order confirmed that the observed differences are not a result of a different sequencing coverage between samples.

In order to assess whether particular genomic features exhibited different methylation levels in mESC-derived neurons, we measured the weighted methylation levels (mCG/CG or mCH/CH) in intergenic regions, introns, exons, CpG islands (CGI), and 500 bp upstream and downstream of transcription start sites (TSS) (Figure 5B and Figure 6B). This revealed genome-wide CG hypermethylation in mESC-derived neurons (~6–12% higher than that of in vivo neurons, absolute methylation level difference), suggesting generalised dysregulation of CG methylation levels. Increased mCH levels were observed in all genomic features except exons. The reason for the specific exclusion of exons from hypermethylation in the mCH context is unknown but suggests that specialised mechanisms regulating exonic mCH levels are conserved in vitro.

Next, we generated ranked lists of genes based on either similarity or difference in genic methylation between neuronal populations and performed gene set enrichment analysis (GSEA) to examine the correlation with biologically relevant pathways. We composed a pathway package by combining gene ontology as well as reactome pathway datasets and performed GSEA on both gene lists. Initially, we performed GSEA based on differences in methylation levels (Figure 5C and Figure 6C). For mCG, the top pathways with higher gene body methylation in vitro represented neuronal activity and synapse formation. Due to the generalised hypermethylation in the CG context, relatively few pathways were enriched for genes with lower methylation values (at *p* < 0.05, Figure 5C), none of which directly related to neurons. The full details of the top 50 hyper- and hypomethylated pathways for mCG context are listed in Appendix A.

For mCH, the top pathways enriched for genes hypermethylated for in vitro neurons included morphogenesis and development, including brain and central nervous system development. However, unlike mCG, differences in neuronal activity and synapse formation were not detected. Genes hypomethylated in the mCH context within in vitro neurons included pathways for cell cycle, cell division, and DNA repair, but again, no pathways relating directly to neurons were identified (Figure 6C). The full details for the top 50 hyper- and hypomethylated pathways for the mCH context are listed in Appendix A.

To determine regional correlation in mCG and mCH levels between mESC-derived neurons, the fetal frontal cortex, and 7-week-old mouse PFC neurons and glia, we performed hierarchical clustering on 100 kb bins of the whole genome (excluding chromosomes X and Y) (Figure 5D and Figure 6D). For mCG, the fetal brain and adult neurons were the most similar, with adult glia joining at the next node, while mESC-derived neurons formed their own branch. The low correlation between neuronal samples is likely due to the overall higher methylation of CG in ESC-derived neurons. For mCH, there was a high similarity between in vitro- and in vivo-derived neuronal datasets, while glia were similar to the fetal sample, which is consistent with the neuron-specific accumulation of mCH. This clustering based on 100 kb bins showed that differences in methylation between neurons were not evenly distributed throughout the genome but showed regional variability.

To further investigate the difference in methylation between in vivo and in vitro neurons, we analysed the genomic distribution of differentially methylated regions (DMRs) between in vitro and in vivo neurons for mCG (Figure 5E) and mCH (Figure 6E). In total, a similar number of DMRs were identified for mCG (34,714) and mCH (28,912). For mCG, DMRs were distributed across all genomic elements including gene bodies, intergenic regions, exons, introns, CGIs, and TSSs. The vast majority of these DMRs showed increased methylation in vitro relative to in vivo consistent with the observed hypermethylation of CG (Figure 5A). The greatest enrichment in DMRs was found in exons and the lowest in intergenic regions. In contrast to mCG DMRs, mCH DMRs showed a more restricted distribution, with enrichment in introns, exons, and intergenic regions, while remaining undetectable in CGIs or TSSs. Furthermore, mCH DMRs showed both hyper- and hypo-methylation for in vitro neurons compared to in vivo neurons, suggesting more complex regulatory differences. To determine whether DMRs could be related to differences in neuronal identity or maturation of the in vitro neurons, we used previously published datasets of mouse embryonic or adult brain cis-regulatory elements [60] as well as developmental DMRs [13]. A small increase in mCG DMRs in the cortical and brain enhancers was observed relative to the total intergenic regions, suggesting potential regulatory differences between in vitro and in vivo neurons. The developmental DMRs fall into two categories: fetal DMRs are those that are methylated in the fetal brain and lose methylation during maturation, while adult DMRs have low methylation in fetal samples but are more highly methylated in adulthood. Our data for the CG context showed an enrichment for fetal DMRs (5.4×) and a depletion for adult DMRs (0.4×). This indicates that while a large number of fetal DMRs become methylated in ESC-derived neurons, most adult DMRs do not change methylation level and retain a methylation state similar to fetal brain samples. As the fetal brain lacks CH methylation, all developmental DMRs for the CH context are adult DMRs, and ESC-derived neurons showed methylation for those DMRs in about two thirds of the locations, while approximately one third remain in a fetal-like lower methylation state. These data suggest that in vitro neurons cannot properly demethylate regions that normally are losing methylation over maturation, which might contribute to the overall hypermethylation observed.

### 3.6. Similarities in Gene Regulation between In Vivo and In Vitro Neurons

To simultaneously identify gene sets that share similar methylation states in both in vitro and in vivo neuron populations and discriminate them from fetal or glial cells, we applied GSEA on genes ranked by a combination of similarity in DNA methylation level between the mESC-derived in vitro neurons and in vivo neurons as well as the dissimilarity to non-neuronal cell types (glia and fetal brain cells, Figure 7A–D).

This analysis was performed for mCG and mCH independently and resulted in an enrichment for pathways linked to genes that have an equivalent methylation state in both of the neuronal samples. It is important to note that this analysis enriches for similarity irrespective of overall methylation levels, whether high or low. In the analysis of mCG, the most represented pathways belonged to two major groups: neuronal function or cell cycle. While neuronal terms are expected to be shared between neurons, the enrichment of the latter group is likely due to its associated genes having a different methylation state in postmitotic neurons as compared to actively proliferating cells. Interestingly, for mCH, with the exception of Wnt signalling, the most highly enriched pathways did not relate directly to neurons but included pathways related to chromatin organization, transcription, and splicing (Figure 7 and Appendix A). Together with the observed differential localisation of mCG and mCA in neuronal nuclei, these data suggest that rather than being directly involved in neuronal specification, mCH could play a role in the dynamic reorganisation of chromatin and the regulation of alternative splicing that occurs during neuronal maturation [61,62].

We next analysed the correlation between gene body mCH levels and gene expression for in vitro neurons and compared this to the adult cortical neurons from Lister et al. [13]. This analysis confirmed an inverse relationship, which was consistent with the observation in cortical neurons (Figure 7E). In agreement with this, the comparison of RNA-seq data from the in vitro neurons relative to published ChIP-seq datasets for H3K27me3 (a repressive histone modification) [63] and H3K36me3 (an active histone modification) [64], suggested that mCH methylation levels are inversely correlated to active chromatin and positively correlate to repressive chromatin (Appendix A). Interestingly, we observed an enrichment for H3K27me3 in genes with the highest increase in methylation when comparing in vitro neurons to in vivo neurons. These genes still harbor relatively low levels of methylation in the CG context in both neuronal populations; it has been proposed that polycomb-associated H3K27me3 can partly compensate for mCG, and thus these genes might be regulated by this mechanism [65].

To investigate the relationship between methylation and gene expression of the in vitro neurons more closely, we analysed cortical enhancer mCG methylation levels in vivo and in vitro using a published cortical enhancer dataset [60]. Interestingly, this analysis revealed that while most enhancers showed no significant CG methylation differences between the in vivo and in vitro neurons, a small subset of cortical enhancers were hyper-methylated in vitro (<4% ΔmCG > 0.5, Figure 7F). To investigate these more closely, we selected the most hypermethylated enhancers in vitro and found that GO terms for genes linked to those enhancers were enriched for synaptic processes, cell differentiation, and RNA processing (Appendix A). Furthermore, the genes for a subset of splicing factors (CUG-BP, Elav-like (Celf) family) with important roles in neurodevelopment [66] were also in close proximity to multiple enhancers. We analysed the expression and gene body mCH methylation for transcription factors (TFs) with binding motifs within these enhancers (Figure 7F–H). This analysis revealed that enhancer mCG hypermethylation most commonly correlated with lower or no TF gene expression in vitro and TF gene body mCH hypermethylation. However, exceptions to this were identified with several genes (Rfx3, Tfap4, and Npas1) showing an inverse relationship (higher enhancer CG methylation, lower gene body CH methylation, and higher transcript abundance in vitro). Taken together, these data suggest that differences in transcription factor abundance in vitro can affect the methylation state of cis-acting regulatory elements.

## 4. Discussion

The complex dynamics, composition, and patterns of DNA methylation observed in the development and maturation of postnatal neurons are hypothesized to play an important role in modifying gene expression and consolidating both neuronal cell types and their response to activity [1,2,4,5,6,20]. Mouse studies have identified some factors involved in the process, including the DNA methyltransferases catalysing the deposition of mC [11,22] and methylation readers such as MeCP2 that link DNA methylation to gene expression [6,9,16,67,68,69,70]. However, many basic questions remain unanswered, including the roles and regulation of dynamic methylation events and the factors that define the targeting of methylation sites. Developing a robust in vitro model system to recapitulate the diverse methylation events particular to neurons is important to facilitating the detailed molecular dissection of these processes. In the present study, we have extended an established protocol to differentiate cortical neurons from mESCs and have shown that these cells acquire in vivo levels of non-CG methylation in a similar time frame to that of in vivo brain development. Furthermore, we have shown that the timing of mCH deposition in vitro correlates to a transient increase in Dnmt3a expression, which also recapitulates that observed in vivo [13]. If, as these results suggest, the deposition of mCH and Dnmt3a expression is indeed hardwired into the developmental process, this has profound implications for studying the equivalent processes in human iPSC-derived neurons. The human brain has a much more extended developmental timeline than does the mouse brain [71], and maximal in vivo mCH methylation levels are not observed until late adolescence (16+ year) [13]. Indeed, mCH levels in human ESC-derived 2-D and 3-D cultured neurons harboured negligible mCH levels (Figure 4). The most advanced in vitro human cerebral organoid differentiation protocols currently available can only recapitulate relatively early embryonic developmental stages (reviewed [72]). Whether further development of human cerebral organoids through in vitro vascularisation [73] or transplantation into the mouse [74] can overcome this developmental obstacle and accelerate human neuron maturation to experimentally tractable time scales is currently unknown. It has been suggested that differences in protein stability or biochemical reaction speed dictate differences in timing between species, which in turn defines the pace of maturation [75,76,77,78], but even with extended culture times, human neurons seem to lack signs of mCH accumulation. Cerebral organoids cultivated for approximately 2 years were shown to recapitulate early postnatal features, but to what extent this maturation is represented in DNA methylation is unknown [79]. To our knowledge, the mouse data presented here are the first reported for mammalian neurons derived in vitro from pluripotent cells that harbour mCH at levels similar to those present in neurons in vivo, opening the door to further, targeted investigations of the regulatory pathways and environmental factors involved.

Detailed analysis of DNA methylation levels and genomic distribution in mouse ESC-derived neurons identified several interesting features. Firstly, the levels and distribution of mCG and mCH were regulated independently. Compared to the mouse brain neurons, generalised hypermethylation in the mCG context, both within gene bodies and across the genome, was not observed for mCH, which showed relatively normal gene body methylation levels and slightly lower exon methylation. Regions of higher methylation in the CG context were not generally accompanied by higher methylation in the CH context and vice versa, and regions showing comparably lower methylation in one context did not show the same pattern in the other context. These observations suggest that different regulatory mechanisms are involved in the remodelling of neuron methylation patterns during maturation, depending upon the DNA context and genomic feature targeted. As the mESC-derived neurons did not recapitulate either of these methylation contexts with complete fidelity, it is likely that other factors, such as environment, neuronal connectivity, and synaptic activity, all act to develop the mature neuronal methylome. Similarly, our comparisons to previously published iN cell data [55] suggest that in this model system, other factors are also required to fully develop in vivo methylation levels and patterns.

The early postnatal nuclear landscape of neurons is highly dynamic, with changes in gene transcription [80], alternative splicing [62,81], DNA methylation [12,13,17], and chromatin remodelling [82]. It is likely that the regulation of all these facets of neuron development are tightly interconnected. It is well-established that gene body mCH levels in neurons is inversely correlated with gene expression [13,54,67,83], which is consistent with the findings shown here, and it has been suggested that establishment of early postnatal mCH regulates the transcription of affected genes at later time points [6]. Indeed, we have shown that changes in gene body mCH methylation are associated with both higher and lower transcript abundance. Immunologically labelled mCA, both in vivo and in vitro, shows a strong association with the nuclear periphery, suggesting association of these genomic regions with the nuclear lamina. As the nuclear lamina forms a repressive environment for transcription [84], association and dissociation of genes with this environment can be a potent regulator of expression. During the neural induction of mESCs, the proneural gene MASH1 is translocated away from the nuclear lamina, concomitant with upregulation of its expression [85]. Similarly, hundreds of genes change lamina interactions during differentiation from mESC to neural progenitor cells and subsequently to astrocytes, and genes affected by altered lamina interactions reflect cell identity and influence the likelihood of a gene being subsequently activated [86]. These findings suggest that lamina–genome interactions are centrally involved in the control of gene expression programs during lineage commitment and terminal differentiation. The association and dissociation of genes from the nuclear lamina is not restricted to developing or differentiating cells. For example, the BDNF gene is translocated away from the nuclear lamina, with a concomitant increase in expression following stimulation of mature neurons in vivo, proving that transcription-associated gene repositioning can occur in adult neurons as a result of enhanced activity [87]. While it is not yet known how mCA associates with the nuclear lamina or whether this is a direct association, one possibility could involve binding to MeCP2 [6,13,67,83]. MeCP2 is a multifunctional protein, with reported roles in both the repression and upregulation of gene expression, as well as involvement in nuclear structure [69,88,89]. In addition to binding various methylated DNA species through its methyl-binding domain, including mCA, mCG, and 5hmC [11,67,69,83], MeCP2 is able to interact directly with the lamin-B receptor [90], a role that is independent of its function as an epigenetic reader protein. As levels of MeCP2 are very high in neurons and approach H1 linker histone levels [70,91], it is tempting to speculate that one role of MeCP2 binding to mCA is to regulate its association with the nuclear lamina. The level of MeCP2 has been shown to increase during mESC-derived neuron differentiation [92], which is consistent with a close regulatory association to the increased levels of mCA observed here.

Generation of base resolution DNA methylomes of mESC-derived neurons allowed us to compare pathways for which gene sets showed either the greatest similarity or the greatest difference in methylation levels to in vivo neurons. Pathways hypermethylated in the mCH context in ESC-derived neurons relative to in vivo neurons included a range of broad developmental pathways, with no particular emphasis on neuron maturation or function. As mCH has been shown to be deposited in the bodies of lowly expressed genes [6], this suggests that these pathways have reduced functionality in postmitotic ESC-derived neurons. In contrast, pathways with hypomethylation in the mCH context represented the cell cycle and mitosis. These pathways are predicted to be silenced in postmitotic cells, suggesting that this hypomethylation occurs as a result of the inclusion of the represented genes in tightly packed heterochromatin, although further studies are needed to confirm this. Taken together, these data suggest that despite the different levels and labelling patterns for mCH in the mESC-derived neurons, some fidelity is retained in the ESC-derived cells. Interestingly, gene set enrichment pathways showing the greatest similarity in mCH between in vivo and ESC-derived neurons include a range of regulatory pathways involved in transcription, splicing, and chromatin organisation, all aspects of neuron development that are significantly modified in early postnatal neurons. Whether mCH is involved in the regulation of alternative splicing or chromatin remodelling in neurons remains to be determined. However, direct links between mCA and alternative splicing have been shown in human ESCs [93], and CG methylation contributes to the inclusion or exclusion of alternatively spliced exons in human cell lines [94]. Furthermore, as MeCP2 can also be associated with alternative splicing and nuclear architecture [89,92], a potential role for mCH in these processes should be considered.

The generalised CG hypermethylation in mESC-derived neurons challenged the analysis of gene sets enriched for either similarity or difference to in vivo neurons, as the highest ranked gene sets for both analyses represented different but closely related aspects of neuron development and function. The mechanism underlying mCG hypermethylation is not yet known. It could reflect either increased methylation, decreased activity of demethylation pathways [95], an accumulation of 5hmC [3,17], or a combination of any or all three. We observed that many developmental DMRs and cortical enhancers stay methylated during in vitro differentiation, which might suggest that the inability to demethylate these regions contributes to the overall hypermethylation. The genome-wide increase in mCG was observed to occur earlier in the neuron differentiation timeline than the maximal increase in mCH, suggesting that these two processes are regulated independently. It is also possible that early changes in mCG levels could in part underlie the differences observed in mCH methylation.

## 5. Conclusions

Despite several differences in DNA methylation and gene expression patterns that indicate that in vitro neurons are not fully matured, important hallmarks of DNA methylation, especially for mCH, are present in the in vitro system presented here. mCH is anticorrelated to gene expression and agrees with the presence or absence of major histone marks. Furthermore, intranuclear localisation of mCA close to the nuclear lamina was found in neurons of the brain as well as in cell culture-generated neurons. As a whole, this work establishes that in vitro differentiation of mouse embryonic cells to neurons is a highly tractable and valuable model system with which to further dissect the roles of DNA methylation and higher order intranuclear architecture on neuron maturation and function. Moreover, insights gained from this system may in turn permit rational manipulation of these epigenetic regulatory programs in human iPSC-derived neurons in order to more closely resemble that of mature human neurons.

## Figures and Tables

**Figure 1 genes-14-00957-f001:**
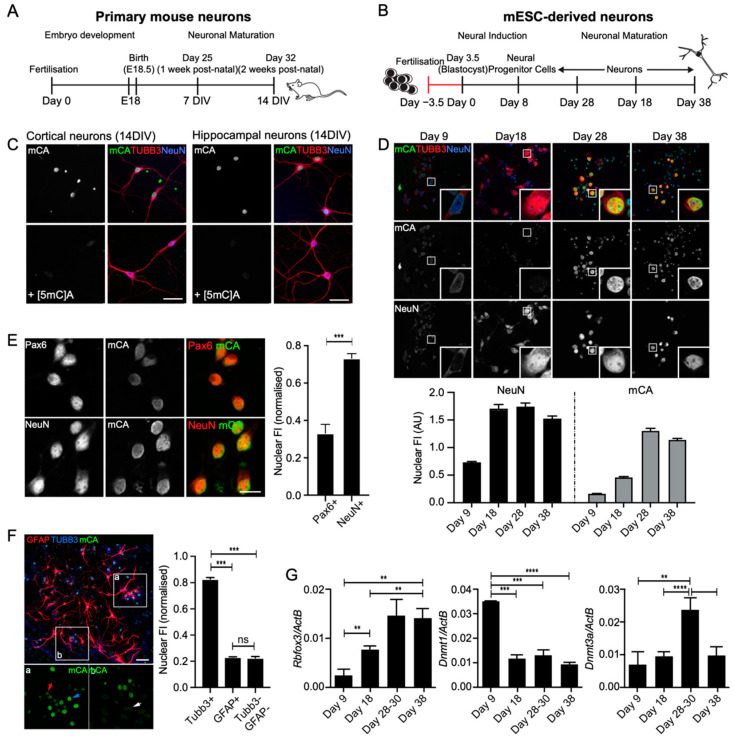
In vitro acquisition of DNA methylation in primary and ESC-derived mouse neurons. (**A**,**B**) Timeline schematic of neuron development in the mouse cortex (**A**) and from mESCs in vitro (**B**). (**C**) 14 DIV cortical or hippocampal neurons immunolabelled for NeuN, TUBB3, and mCA ± 2.5 µM [5mC]A. Scale bar = 50 µm. See also Appendix A. (**D**) mESC-derived neurons fixed at different times during maturation and labelled for NeuN, mCA, and DAPI. Scale bar = 20 µm. The nuclear fluorescence intensity (FI, Arbitrary Units) of NeuN and mCA was determined. Results = mean ± SEM for one representative differentiation. Similar temporal labelling profiles were observed in 3 separate differentiations. (**E**) Neural progenitors (Day 9–10, Pax6+) and neurons (Day 20–40 NeuN+) were immunolabeled for mCA, and the level of mCA nuclear fluorescence intensity was quantified. Individual differentiation data were pooled and normalised. Results show mean ± SD, with n = 3 differentiations. Scale bar = 10 µm. (**F**) Later-stage neural differentiations (Day 30–38) were immunolabeled for astrocytes (GFAP+), neurons (TUBB3+), and mCA. Nuclei were manually masked, and the level of mCA measured in TUBB3+, GFAP+, and GFAP-/TUBB3- cells. Data were pooled and normalised. Results show mean ± SD, with n = 2 independent differentiations. Scale bar = 50 µm. For each nuclei quantitation between 50 and 300, individual nuclei were analysed. See also Appendix A. (**G**) Gene expression of Dnmt1, Dnmt3a, and Rbfox3 (NeuN) determined using RT-qPCR relative to β-actin. Results = mean ± SEM, with n = 3–4. For all experiments, statistical analysis was performed using an unpaired Student’s *t*-test. ** *p* < 0.05, *** *p* < 0.001, **** *p* < 0.0001, ns = not significant.

**Figure 2 genes-14-00957-f002:**
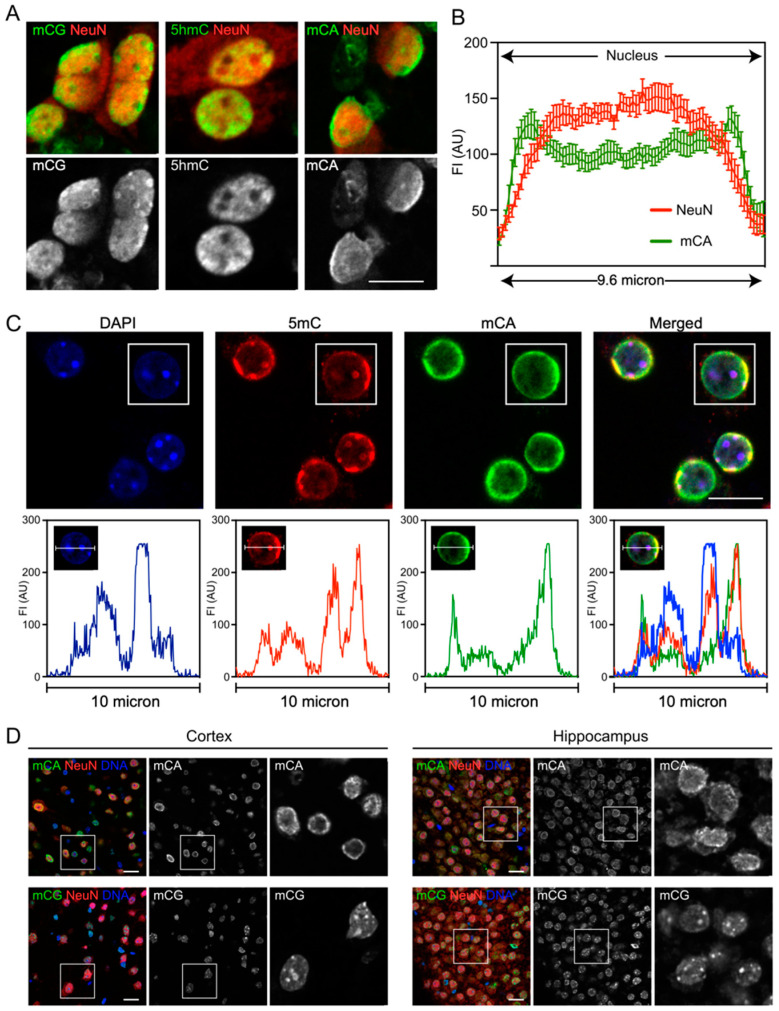
Intranuclear labelling of DNA methylated in different sequence contexts. (**A**) mESC-derived neurons were immunolabelled for NeuN and either mCG, mCA, or 5hmC. (**B**) Fluorescence intensity through the nuclei of mESC-derived neurons labelled for mCA and NeuN is shown using line scans. Results show the mean intensity ± SEM, with n = 10 nuclei. Scale bar = 10 µm. (**C**) mESC-derived neurons were immunolabelled for total 5mC (mCG + mCH), mCA, and NeuN, and nuclei identified using DAPI. The distribution of methylation labelling is shown using line scans (representative nucleus). Scale bar = 10 µm. (**D**) Immunohistochemical analysis of DNA methylation in adult mouse hippocampus and cortex immunolabeled for either mCG or mCA and NeuN. Nuclei identified using YOYO1. Scale bar = 20 µm.

**Figure 3 genes-14-00957-f003:**
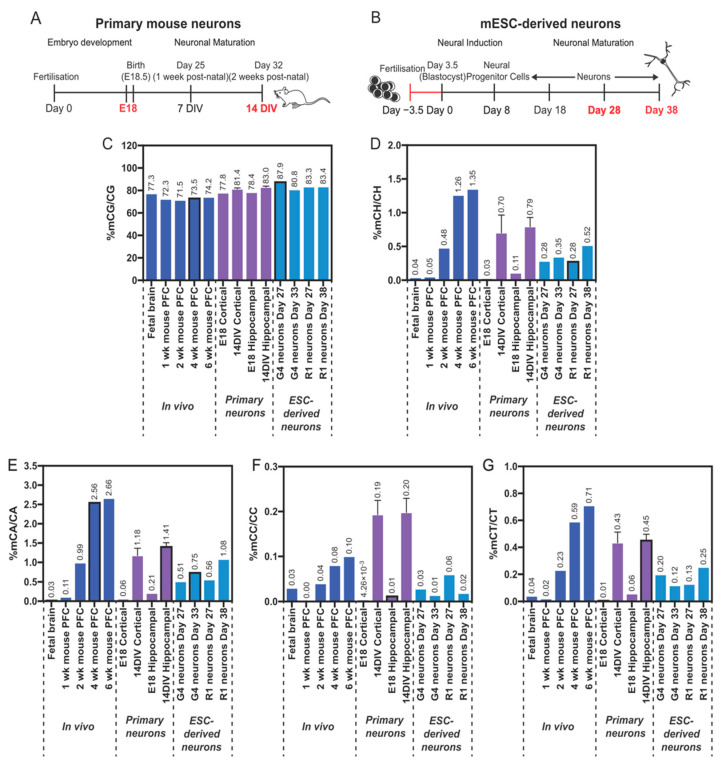
WGBS analysis of DNA methylation in primary mouse neurons and mESC-derived neurons. (**A**) Timeline schematic of neuron development in the mouse cortex. (**B**) Timeline schematic of neuron development from mESCs in vitro. Time points analysed with WGBS are highlighted in red. (**C**–**G**) Global levels of DNA methylation in the mouse brain ((13), dark blue bars), primary mouse cortical or hippocampal neuron cultures (E18 and 14 DIV, purple bars), and mESC-derived neuronal cultures (R1 and G4 cell lines, light blue bars). Values represent the weighted DNA methylation levels: the fraction of all WGBS base calls that were C at cytosine positions in the genome (for each context separately). Results = single samples, except 14 DIV where n = 2, mean ± SD.

**Figure 4 genes-14-00957-f004:**
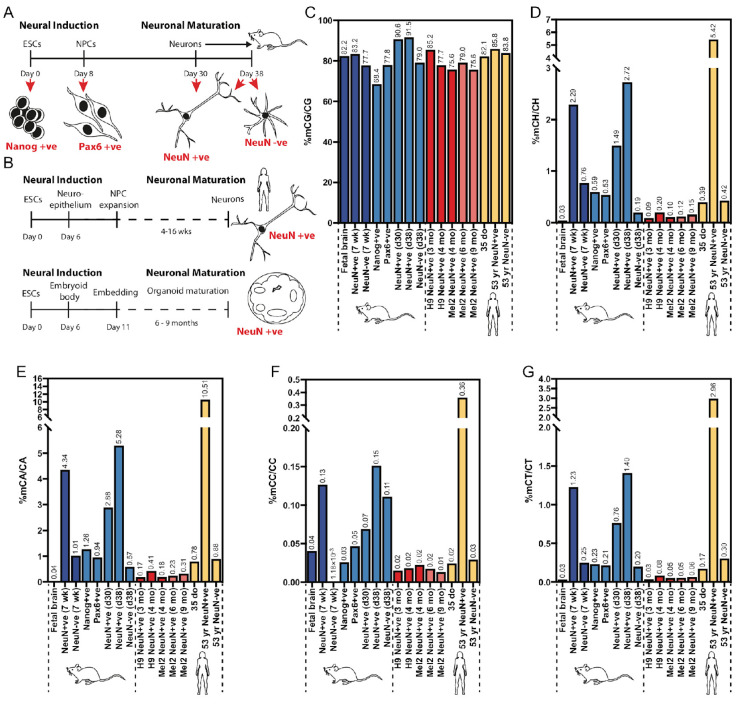
DNA methylation levels in fluorescence-activated sorted nuclei. (**A**,**B**) Schematics showing time points during the mESC and hESC differentiation to neurons at which nuclei were isolated. (**A**) Samples from the mESC differentiation included Nanog +ve mESCs, Pax6 +ve neural progenitors, day 30 and 38 NeuN +ve mouse neurons, and day 38 NeuN −ve cells. (**B**) In the human ESC differentiation, NeuN +ve nuclei were isolated following 12–16 weeks of 2-D culture or 6–9 months of 3-D culture. (**C**–**G**) Level of DNA methylation in isolated nuclear populations. The light blue (mouse), red (human 2-D), and orange (human 3-D) bars show samples generated in this study. The dark blue (mouse) and yellow (human) bars show previously published levels of DNA methylation [13].

**Figure 5 genes-14-00957-f005:**
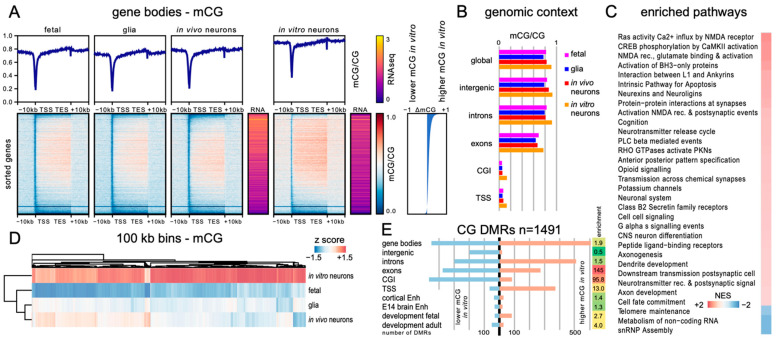
Global mCG properties of in vivo versus in vitro generated neurons. DNA methylation levels in fluorescence-activated sorted nuclei. mCG characteristics for d38 mESC-derived neurons (in vitro neurons), 7-week-old mouse prefrontal cortex neurons (in vivo neurons), NeuN-negative cells from 7-week-old mouse prefrontal cortex (glia), and fetal mouse frontal cortex (fetal). (**A**) Weighted CG DNA methylation levels (mCG/CG) for all genes and 10 kb flanking regions are shown at the top. The main heatmap (red/blue) shows the mCG of genes and flanking regions sorted by the difference in gene body mCG/CG between mESC-derived neurons and in vivo adult mouse PFC neurons, as indicated on the very right of the heatmap. The normalized TPM values for each gene from RNA-seq data of neurons are shown in the smaller heatmap (gold/purple). (**B**) Weighted methylation levels (mCG/CG) for the whole genome, intergenic regions, introns, exons, CpG islands (CGI), and 500 bp flanking transcription start sites (TSS). (**C**) Enriched pathways after preranked gene set enrichment analysis based on differences in gene body mCG/CG between in vivo and in vitro neurons. The top pathways based on enrichment scores (NES) for genes with higher mCG/CG in vitro (positive NES score) and lower mCG/CG in vitro (negative NES score) are shown. (**D**) Hierarchical clustering based on Spearman correlation of mCG levels in all 100 kb bins of the genome. (E) Location of differentially methylated regions (DMRs) between in vivo and in vitro neurons. Regions with higher mCG in vitro are shown in red, and regions with lower mCG in vitro are shown in blue. The enrichment score was calculated based on the observed numbers of DMRs per context compared to the random distribution of DMRs across the genome.

**Figure 6 genes-14-00957-f006:**
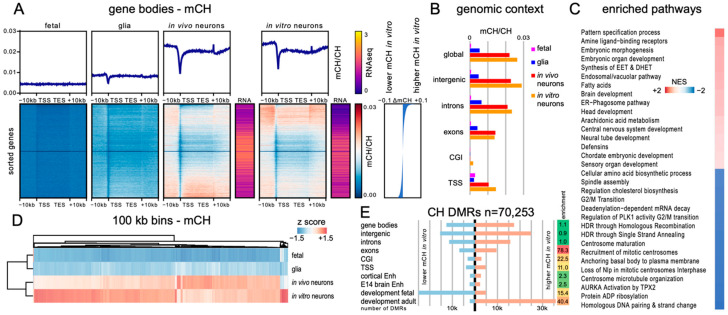
Global mCH properties of in vivo versus in vitro generated neurons. mCH characteristics for d38 mESC-derived neurons (in vitro neurons), 7-week-old mouse prefrontal cortex neurons (in vivo neurons), NeuN-negative cells from 7-week-old mouse prefrontal cortex (glia), and fetal mouse frontal cortex (fetal). (**A**) Weighted CH DNA methylation levels (mCH/CH) for all genes and 10 kb flanking regions are shown at the top. The main heatmap (red/blue) shows the mCH of genes and flanking regions sorted by difference in gene body mCH/CH between mESC-derived neurons and in vivo adult mouse PFC neurons, as indicated on the very right of the heatmap. Normalized TPM values for each gene from RNA-seq data of neurons are shown in the smaller heatmap (gold/purple). (**B**) Weighted methylation levels (mCH/CH) for the whole genome, intergenic regions, introns, exons, CpG islands (CGI), and 500 bp flanking transcription start sites (TSSs). (**C**) Enriched pathways after preranked gene set enrichment analysis based on differences in gene body mCH between in vivo and in vitro neurons. The top pathways based on enrichment scores (NES) for genes with higher mCH/CH in vitro (positive NES score) and lower mCH/CH in vitro (negative NES score) are shown. (**D**) Hierarchical clustering based on Spearman correlation of mCH levels in all 100 kb bins of the genome. (**E**) Location of differentially methylated regions (DMRs) between in vivo and in vitro neurons. Regions with higher mCH in vitro are shown in red, and regions with lower mCH in vitro are shown in blue. The enrichment score was calculated based on the observed numbers of DMRs per context compared to the random distribution of the DMRs across the genome.

**Figure 7 genes-14-00957-f007:**
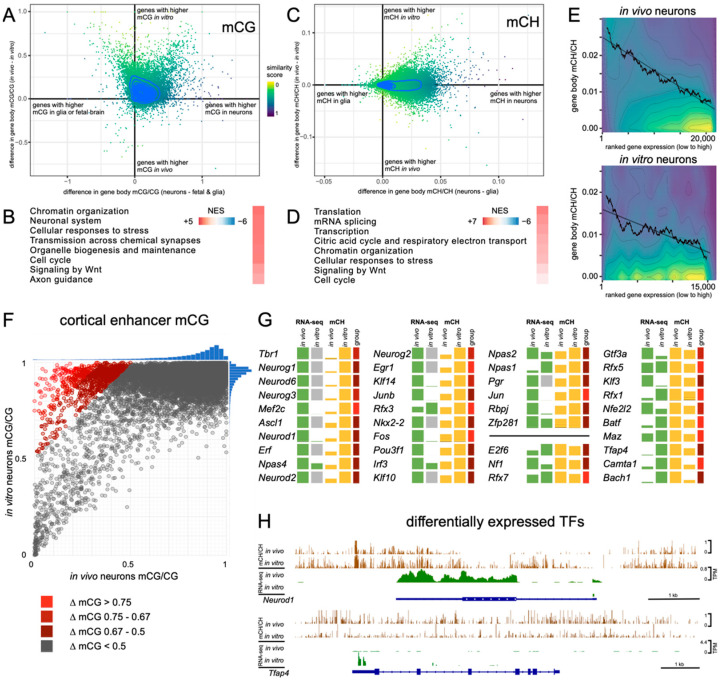
Enrichment for genes with similar methylation patterns for in vitro neurons and for in vivo neurons. (**A**) Differences in gene body mCG level between neurons and fetal brain as well as glia (*x*-axis), and differences in gene body mCG level between both neuronal populations (*y*-axis) for all 24,049 genes with WGBS coverage. Dots are coloured by the similarity score that was used for gene set enrichment. The blue contour plot is based on gene density. (**B**) Selection of the top enriched pathways from GSEA using similarity scoring from mCG information. (**C**) Same as (**A**) but for the mCH context. (**D**) Selection of the top enriched pathways from GSEA using similarity scoring from mCH information. Full details for the top 20 similarity-enriched pathways are listed in Appendix A. (**E**) The anticorrelation between CH methylation in gene bodies (*y*-axis) and gene expression (*x*-axis) was preserved for the in vitro neurons. The sliding window-averaging over 20 genes and the fitted line over a density plot are shown. (**F**) mCG within 22,621 cortical enhancers from Shen et al. [60] for in vivo neurons (*x*-axis) and in vitro neurons (*y*-axis). The colour indicates the methylation difference between both neuronal populations for specific enhancers. (**G**) The expression and gene body CH methylation state for the transcription factors with enriched binding motifs in enhancers shown in (**F**). Gene expression based on TPM values from RNA-seq and mCH are normalised per gene to the higher of both values from the in vivo and in vitro neurons. A grey box indicates that no reads were found in RNA-seq data for the specific gene. Genes are ordered for differences in gene body mCH. Group colours are the same as used in (**F**). (**H**) Example gene browser shots for Neurod1 (higher methylated in vitro, more expressed in vivo) and Tfap4 (higher methylated in vivo, more expressed in vitro). mCH is shown in brown, TPM-normalized gene expression is shown in green, and the Ensembl gene location is shown in blue.

## Data Availability

The datasets supporting the conclusions of this article are available in the GEO repository under accession number GSE137098: https://www.ncbi.nlm.nih.gov/geo/query/acc.cgi?acc=GSE137098 accessed on 1 April 2023. Reviewers, please use the token “qdajswsylbgvxun” to access data. A genome browser session showing DNA methylation and RNA expression for in vitro and in vivo neurons is accessible at https://tinyurl.com/y2gmmb4b accessed on 1 April 2023.

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
