# Peer review of "Embryonic Stem Cell-Derived Neurons as a Model System for Epigenome Maturation during Development"

_genes, 2023, doi:10.3390/genes14050957_

Round 1

Reviewer 1 Report

Major points:

  1. The detailed base resolution DNA methylomes in Fig 5-7 is unquestionably great, but many will have an enormous curiosity right after Fig 4 on why the similar in vitro neuronal differentiation strategy just can not work on the human system on recapitulation of mCH phenotypes. People can argue for the difference in protocol used fro differentiation or (as mentioned in the discussion) the species-specific difference in biochemical reaction rate. However, since it has been proposed that Dnmt3a is responsible for methylation of CH sites in the mouse system. It would be great to check whether 1) in the human in vitro neuronal cultures (either the 2D culture or the organoid), DNMT3A was expressed or not at the time point examined and 2) over-expression of DNMT3A itself was sufficient to induce mCH, at least to some level. As a complement, knockdown of Dnmt3a in the mouse in vitro neuronal cultures and see whether the mCH could be abolished may provide another piece of evidence to support the claim. 

Minor points:

  1. Fig 2c, It will be better to present mCG figure side by side with mCA figure, but not the 5mC figure. Since the difference between subnuclear location of mCG and mCA is the main point for comparison. Also both Fig 2A and 2D went such way, this change will make the whole figure more consistent.
  2. Fig 1, the blue channel is really dim for visualization, maybe can consider using cyan?

Reviewer 2 Report

Authors has nicely correlated the level of mCH methylation in mESC derived neuronal population with that of in-vivo conditions. The proposed in-vitro model will have a great significance in elucidating mechanism underlying various neurological diseases.

The manuscript provides a novel model and valuable model system to dissect the roles of DNA methylation and higher order intranuclear architecture on neuron maturation and function. However, there are certain corrections,Kindly consider the following:

1.       Materials and Methods

1.       Please specify the place from where the mice for all experimental protocols were procured

2.       Specify the ethical certificate number

3.       How were E18 neural progenitors Isolated?

4.       Kindly expand WGBS (An abbreviation needs to be written in full the first time it appears in the text)

5.       Expand “BS” Line 204 and 205

6.       “With a secondary 204 fluorescently labelled antibody in BS” Line 204 name the secondary antibody

7.       Mention the secondary antibodies in Line 198

8. “Adult C57BL/6 mice (8-10 weeks old) were sacrificed, and brains dissected and snap 102 frozen on dry ice.” Kindly mention if the mice were perfused or not and how were they sacrificed

9. “2.14. Statistical Analysis” This section should be laborated

2.       Result

1.       Mention the Stats for Figure 1D

2.       Scale bars in all pictures is required

3.       Fig2: Kindly include the Graph depicting number of cells positive for methylation and mention how many regions were scanned for all microscopy images

3.       Discussion:

1.       “but even with ex- 830 tended culture times human neurons seem to lack signs of mCH accumulation” Explain why

4.       Minor errors:

·       ml is written as mL

·       in vitro italicize

·       After post-fixation Line 205, remove “after”

·       include the catalogue for e Macherey-Nagel NucleoSpin RNA kit

·       Write the concentration of PBS

“using standard protocols” Line 234 give reference

Kindly find the comments attached herewith.

Author Response

Please see the attachment,
